# A gut-activated NHR-86–CYP pathway mediates the neuroprotective effects of *Enterococcus faecium* probiotics in a nematode model of amyotrophic lateral sclerosis

Yu Sang[1]○, Jie Ren[1]○, Alejandro Aballay[1,2]*

**1** Department of Genetics, The University of Texas MD Anderson Cancer Center, Houston, Texas, United States of America, **2** Department of Microbiology and Molecular Genetics, McGovern Medical School at UTHealth, Houston, Texas, United States of America

319 These authors contributed equally to this work.
* aaballay@mdanderson.org

## Abstract

Neurodegenerative diseases are often associated with oxidative stress, and while probiotics may influence neuronal health, the underlying mechanisms remain poorly understood. Using the *sod-1* A4V^M amyotrophic lateral sclerosis (ALS) model in *Caenorhabditis elegans*, we investigated the protective effects of the probiotic *Enterococcus faecium* against oxidative stress-induced neurodegeneration. Animals fed *E. faecium* showed reduced motor neuron degeneration under oxidative stress compared to those maintained on a standard *Escherichia coli* diet. Transcriptome analysis revealed a significant enrichment of oxidoreductase genes, including cytochrome P450 (*cyp*) genes. RNAi-mediated knockdown of *cyp* genes impaired *E. faecium*-mediated neuroprotection, and this loss correlated with increased reactive oxygen species (ROS) levels. We identified the conserved nuclear hormone receptor NHR-86 as a key regulator of *cyp* gene expression and neuroprotection. Loss of *nhr-86* abolished the probiotic's protective benefits, while transgenic expression of *nhr-86* restored *cyp* induction and neuronal resilience. Importantly, intestinal expression of NHR-86 was sufficient to restore CYP induction and neuronal resilience, whereas neuronal knockdown had no effect, indicating that gut NHR-86 activity is essential for this protective pathway. These findings reveal a previously uncharacterized NHR-CYP regulatory axis activated by an intestinal probiotic, highlighting a mechanistic link between microbial signals and host neuroprotection.

## Introduction

Neurodegenerative diseases, including Alzheimer's, Parkinson's, Huntington's, and amyotrophic lateral sclerosis (ALS), progressively damage the nervous system,

**Data availability statement:** All relevant data are within the paper and its Supporting information files.

**Funding:** This work was fully supported by the National Institutes of Health (NIH) through grants GM0709077 awarded to AA and AI117911 awarded to AA. The funders had no role in study design, data collection and analysis, decision to publish, or preparation of the manuscript. None of the authors received a salary from the funders for this work.

**Competing interests:** The authors have declared that no competing interests exist.

**Abbreviations:** ALS, amyotrophic lateral sclerosis; BHI, brain–heart infusion; DAVID, Database for Annotation, Visualization, and Integrated Discovery; GO, gene ontology; HNF4, hepatocyte nuclear factor 4; LB, Luria-Bertani; NGM, nematode growth medium; NHRs, nuclear hormone receptors; qRT-PCR, quantitative reverse transcription PCR; RNAi, RNA interference; ROS, reactive oxygen species.

leading to severe physical and cognitive impairments. ALS, a fatal neurodegenerative disorder, is characterized by motor neuron degeneration, resulting in muscular atrophy, weakness, dysphagia, dysarthria, dyspnea, limb motor impairments, and eventual paralysis in the advanced stages. Respiratory failure and death usually occur within 2−3 years. Over 40 gene mutations have been linked to ALS [1]. Among these, the most common genetic cause is the chromosome 9 open reading frame 72 (C9orf72) hexanucleotide repeat expansion, followed by mutations in TAR DNA-binding protein and fused in sarcoma [2,3]. Mutations in SOD1 account for ~10%–20% of familial ALS and ~1%–2% of all ALS cases [4]. However, few clinical therapeutic or preventive strategies specifically target SOD-1 [5]. The SOD-1 A4V mutation is the most common SOD-1 mutation in the United States, accounting for about 50% of cases [6]. It is conserved in mammals, was the first identified SOD-1 mutation, and remains a primary cause of ALS among SOD-1-related cases. Here, we investigated the protective role of probiotics in preventing motor neuron degeneration using the *sod-1* A4V$^M$ model, which carries a single copy of SOD-1 A4V mutation and exhibits heightened degeneration due to oxidative stress.

We used *C. elegans* as a powerful model organism for studying neurodegeneration due to its simple and well-characterized nervous system. Its transparent body and short life span allow for real-time observation of neuronal function and degeneration, making it ideal for investigating the mechanisms of neurodegenerative diseases [7]. The ability to introduce human disease mutations, such as those linked to ALS, further enhances its utility in modeling motor neuron degeneration [8].

Accumulating evidence suggests that probiotic bacteria play a vital role in human health and well-being by modulating key physiological processes [9,10]. Among these, probiotics have been shown to enhance resistance to oxidative stress, a major contributor to motor neuron degeneration [11,12]. *Enterococci*, particularly *Enterococcus faecium*, have been widely used as probiotics due to their beneficial effects on host health [13,14]. Our recent studies demonstrate that *E. faecium* effectively protects the host against a lethal *Salmonella enterica* infection by activating the Wnt pathway, a crucial regulator of cellular homeostasis [15]. These findings suggest that *E. faecium* may offer neuroprotective benefits in vivo, potentially mitigating neurodegenerative disorders caused by oxidative damage.

The cytochrome P450 (CYP) family plays a critical role in reducing free radicals and influencing neurodegenerative diseases. While some CYP enzymes contribute to oxidative stress, others detoxify reactive oxygen species (ROS), regulate lipid metabolism, and clear toxic metabolites, making them potential targets for neuroprotective therapies. Growing evidence links CYP expression patterns to brain activity and neurological function. For example, CYP46A1 has been identified as a therapeutic target in Alzheimer's disease, suggesting its potential influence on disease progression [16,17]. In Parkinson's disease, CYPs exhibit both protective and harmful effects, further highlighting the complexity of their roles [18,19]. Overall, CYP-mediated endogenous metabolism is closely connected to neurodegenerative diseases and plays a significant role in the central nervous system, influencing steroid synthesis, inflammation, and drug metabolism.

Nuclear hormone receptors (NHRs) are transcription factors that control essential biological processes upon binding specific exogenous or endogenous ligands. Overexpressing NHRs in the intestine can alleviate motor neuron degeneration, with NHRs mediating the protective effects of propionate against α-synuclein-induced neuronal death in a *C. elegans* model of Parkinson's disease [20]. Moreover, recent studies have demonstrated that NHR activation modulates neuronal survival pathways and mitochondrial homeostasis, highlighting their potential as therapeutic targets for neurodegenerative diseases [21].

Utilizing a single-copy ALS SOD-1 knock-in model in *C. elegans*, we studied the protective role of *E. faecium* against motor neuron degeneration under oxidative stress. We found that exposure to *E. faecium* triggered a broad induction of cytochrome P450 genes. These CYPs contributed to *E. faecium*-mediated neuroprotection by reducing intracellular ROS levels during oxidative stress. We identified NHR-86, a conserved NHR and the homolog of human hepatocyte nuclear factor 4 (HNF4), as a key regulator of this response. NHR-86 protected against motor neuron degeneration by activating CYP genes in response to *E. faecium*. The intestinal function of NHR-86 is required for CYP induction and neuronal resilience, supporting an indirect role for intestinal oxidative stress mitigation in neuronal protection. Together, these findings reveal a novel neuroprotective mechanism involving CYP activation and suggest potential strategies for targeting neurodegenerative diseases.

## Results

### *E. faecium* protects *C. elegans* against motor neuron degeneration under oxidative stress

The SOD-1 A4V mutation is the most prevalent disease-associated SOD-1 allele among ALS patients in North America [6]. Patients carrying this mutation typically experience selective degeneration and death of cholinergic spinal motor neurons, while glutamatergic neurons remain functional. Similarly, in *C. elegans*, the *sod-1* A4V$^M$ model exhibits cholinergic motor neuron loss under oxidative stress, with glutamatergic neurons unaffected. Moreover, single-copy *sod-1* A4V$^M$ knock-in animals in *sod-1(-)* background show significantly greater susceptibility to oxidative stress compared to *sod-1* WT$^M$ knock-in animals in the same genetic background [22], exhibiting an accelerated neurodegenerative phenotype, and providing a reliable model for detecting motor neuron degeneration. Under basal conditions, the single-copy SOD-1 A4V$^M$ model does not show cholinergic motor neuron loss or functional defects in young adult animals, indicating that neurodegeneration in this model is primarily stress-dependent [22]. Therefore, we selected the ALS *sod-1* A4V$^M$ model to investigate the effects of probiotic *E. faecium* on the integrity of cholinergic motor neurons under oxidative stress, which was induced using paraquat [23]. Following paraquat exposure, some *sod-1* A4V$^M$ animals displayed defects in cholinergic motor neurons (Figs 1A and S1). To test whether *E. faecium* protects *C. elegans* against cholinergic motor neuron degeneration under oxidative stress, animals were pretreated with *E. faecium* for 24 hours before being transferred to paraquat plates (Fig 1B). *Escherichia coli*, a bacterial food for laboratory maintenance of *C. elegans*, was used as a control. The cholinergic motor neuron integrity was studied using GFP retention/loss in animals carrying the *unc-17p*::GFP gene, which expresses fluorescent proteins in cholinergic neurons. Less than 10 % of *sod-1* WT$^M$ animals showed motor neuron degeneration when pretreated with either *E. coli* or *E. faecium*, with no significant difference between the two treatments (Figs 1C, S1, and S2). Fifty percent of *sod-1* A4V$^M$ animals showed motor neuron degeneration when pretreated with *E. coli*, while only 20 % of *sod-1* A4V$^M$ animals pretreated with *E. faecium* showed motor neuron degeneration (Figs 1C and S2). Importantly, we found that both UV-killed and heat-killed *E. faecium* failed to provide neuroprotection (S3A and S3B Fig), indicating that live *E. faecium* is required for this protective effect.

We also assessed whether *E. faecium*-mediated neuroprotection promotes survival during oxidative stress. Animals treated with *E. faecium* survived significantly better than those treated with *E. coli* (Fig 1D). Since *E. faecium* protects motor neurons against neurodegeneration, we investigated whether *E. faecium* treatment affects locomotion. Animals were pretreated with *E. faecium* followed by paraquat-induced oxidative stress. *E. faecium* rescued the oxidative stress-induced locomotion defect (Fig 1E). These results indicate that probiotic *E. faecium* preserves cholinergic motor neuron integrity and locomotor function under oxidative stress.

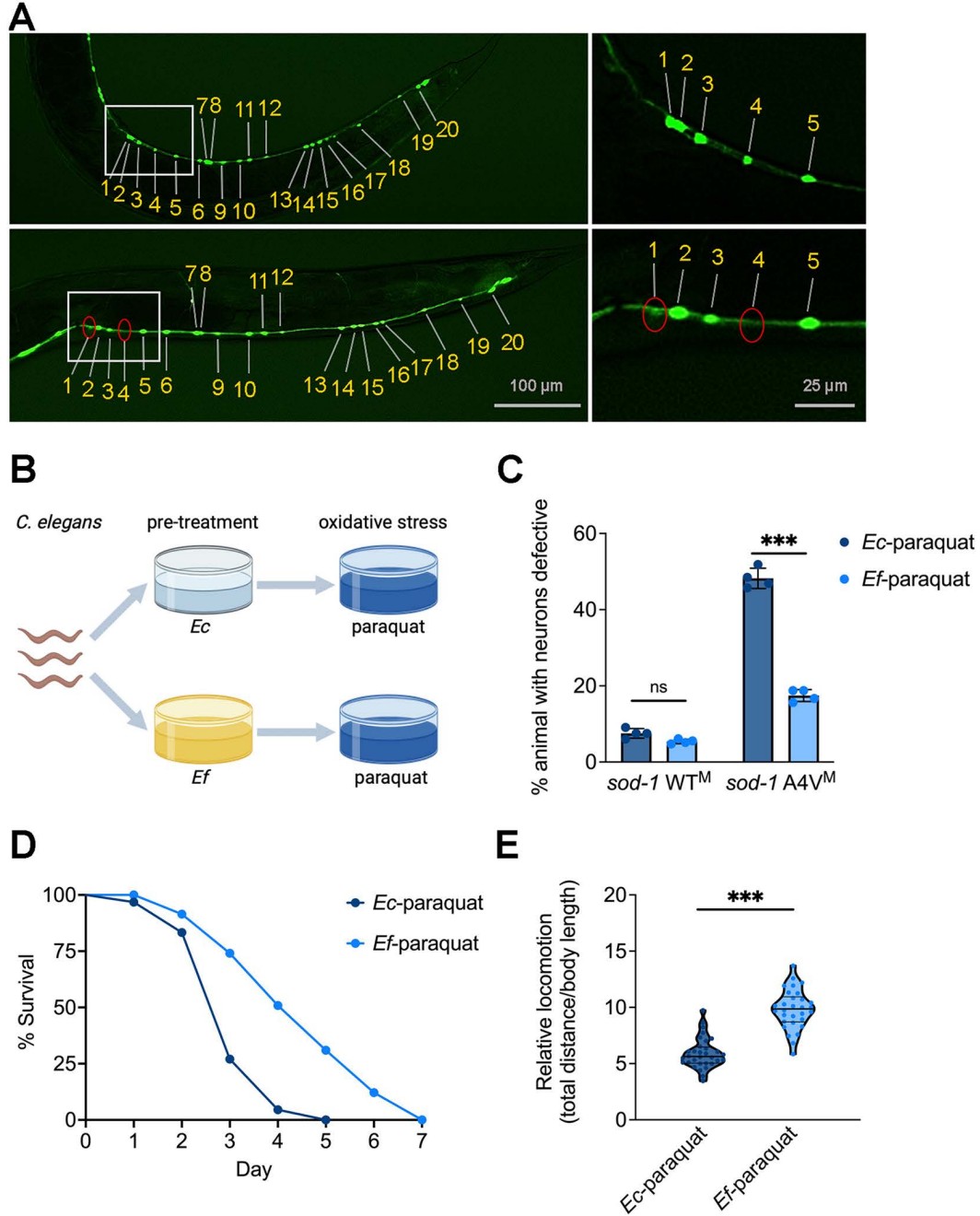

**Fig 1. *Enterococcus faecium* protects against motor neuron degeneration under oxidative stress in *Caenorhabditis elegans*. (A)** Representative images of the ALS *sod-1* A4V^M model show motor neuron degeneration in animals fed *Escherichia coli* followed by oxidative stress induced by 5 mM paraquat. Cholinergic motor neurons were visualized with *unc-17p*::GFP. After 24 hours of exposure, representative images of *sod-1* A4V^M animals were captured, showing those without (top) and with (bottom) neurodegeneration. Enlarged images in the right panels highlight motor neuron defects, indicated by red circles. **(B)** Schematic of the experimental design for probiotic *E. faecium* pretreatment and paraquat exposure. Young adult animals were pretreated on *E. faecium* or *E. coli* lawns for 24 hours, followed by a 24-hour exposure to paraquat. Figure created using BioRender. Sang, Y. (2026) https://BioRender.com/fg7i4lx. **(C)** Oxidative stress induced cholinergic motor neuron degeneration in *sod-1* A4V^M animals. Cholinergic motor neuron integrity was assessed using GFP retention or loss in *sod-1* A4V^M animals following probiotic *E. faecium* pretreatment and paraquat exposure. Cholinergic motor neuron integrity was accessed by scoring for the absence of at least two cholinergic motor neurons posterior to the vulva. **(D)** Survival curve showing *E. faecium*-mediated protection against paraquat-induced oxidative stress. Following 24 hours of paraquat exposure, the survival of *sod-1* A4V^M animals pretreated on *E. faecium* or *E. coli* lawns was recorded. Animals were transferred to new paraquat plates daily until all animals stopped

laying eggs. **(E)** Locomotion of *sod-1* A4V[M] animals following paraquat exposure. Each *sod-1* A4V[M] animal was pretreated on an *E. faecium* or *E. coli* lawn, exposed to 5 mM paraquat for 48 hours, and allowed to move freely for 15 s. Locomotion was quantified using the movement index, calculated as the ratio of movement distance to body length. Columns of defective neurons, survival curves, and locomotion represent assays from three independent experiments (*n* = 15). Abbreviations: *Ec, E. coli* OP50; *Ef, E. faecium*; WT, wild-type. The data underlying this Figure can be found in S1 Data.

Because ALS patients with the SOD1 A4V mutation exhibit loss of glutamatergic neurons, we next examined whether *E. faecium* protects these neurons. Using a *sod-1* G85R[M] model of glutamatergic neurodegeneration [22], we studied the sensory glutamatergic neurons PHA and PHB, which are susceptible to degeneration under paraquat-induced oxidative stress (S4A Fig). *E. faecium* treatment reduced degeneration in these neurons (S4B Fig), indicating that its neuroprotective effects extend beyond cholinergic motor neurons.

## Classic immune pathways are not required in probiotic-mediated neuroprotection

To identify genes and pathways required in *E. faecium*-mediated neuroprotection, we performed transcriptome analysis comparing *E. faecium*-treated and *E. coli*-treated *sod-1* A4V[M] animals. Of the 302 differentially regulated genes, 227 were up-regulated, and 75 were down-regulated upon exposure to *E. faecium* (S1 Table). To identify related gene groups directly or indirectly regulated by *E. faecium* exposure, we performed an unbiased gene enrichment analysis using the Database for Annotation, Visualization, and Integrated Discovery (DAVID; http://david.abcc.ncifcrf.gov/) [24]. The top gene ontology (GO) clusters with the highest DAVID enrichment scores are shown in Fig 2A and 2B and S2 Table. For the down-regulated gene subset responding to *E. faecium* exposure, the most highly enriched clusters were related to innate immune response and transmembrane transport, followed by lipid transport, S-adenosylmethionine biosynthesis, and one-carbon metabolic process (Fig 2A). For the up-regulated gene set, the highest-scoring ontology clusters included oxidoreductase activity and hydrolase activity, followed by genes related to iron ion binding and structure constituent of the cuticle, among others (Fig 2B). As expected, a similar enrichment was also observed using a Wormbase enrichment analysis tool (https://wormbase.org/tools/enrichment/tea/tea.cgi) [25,26], which is specific for *C. elegans* gene data analysis. Innate immune response genes and oxidoreductase activity genes were also highly enriched among down-regulated and up-regulated genes, respectively (S5 Fig and S2 Table). Additionally, we compared our enrichment analysis with a similar study that examined gene expression differences between *E. faecium* and *E. coli* treated wild-type animals. We found comparable results, even though their comparison was between *E. faecium* and heat-killed *E. coli* (S2 Table) [27].

We validated gene expression using qRT-PCR, selecting candidate genes from two key clusters: the innate immune response cluster (top down-regulated) and oxidoreductase activity cluster (top up-regulated). These genes were assessed under four conditions: *sod-1* WT[M] and *sod-1* A4V[M] animals exposed to either *E. coli* or *E. faecium*. Innate immune genes were consistently down-regulated in an *E. faecium*–dependent manner (Fig 2C), while oxidoreductase activity genes were up-regulated (Fig 2D). The qRT-PCR results confirmed the RNA-seq findings, demonstrating a positive correlation and reinforcing that *E. faecium* specifically drives these gene expression changes.

Since the innate immune response cluster was among the most strongly down-regulated by *E. faecium* exposure, we hypothesized that activation of canonical immune pathways may contribute to motor neuron degeneration and that their suppression could be protective. To test this, we used RNAi to knock down classic immune pathways, including PMK-1 [28–30], DAF-16 [31–33], ELT-2 [34,35], and SKN-1 [36,37], in animals subjected to paraquat-induced oxidative stress in the presence and absence of *E. faecium* (Fig 3A). Because RNAi by feeding is largely ineffective in most neurons, these experiments primarily reflect the roles of these genes in the intestine and other non-neuronal tissues. As shown in Figs 3B and S6, RNAi of *pmk-1, daf-16, elt-2,* or *skn-1* did not prevent motor neuron degeneration in animals fed *E. coli* and did not alter the neuroprotective effect of *E. faecium* treatment, indicating that modulation of these canonical immune pathways is not a determining factor in neuroprotection under oxidative stress.

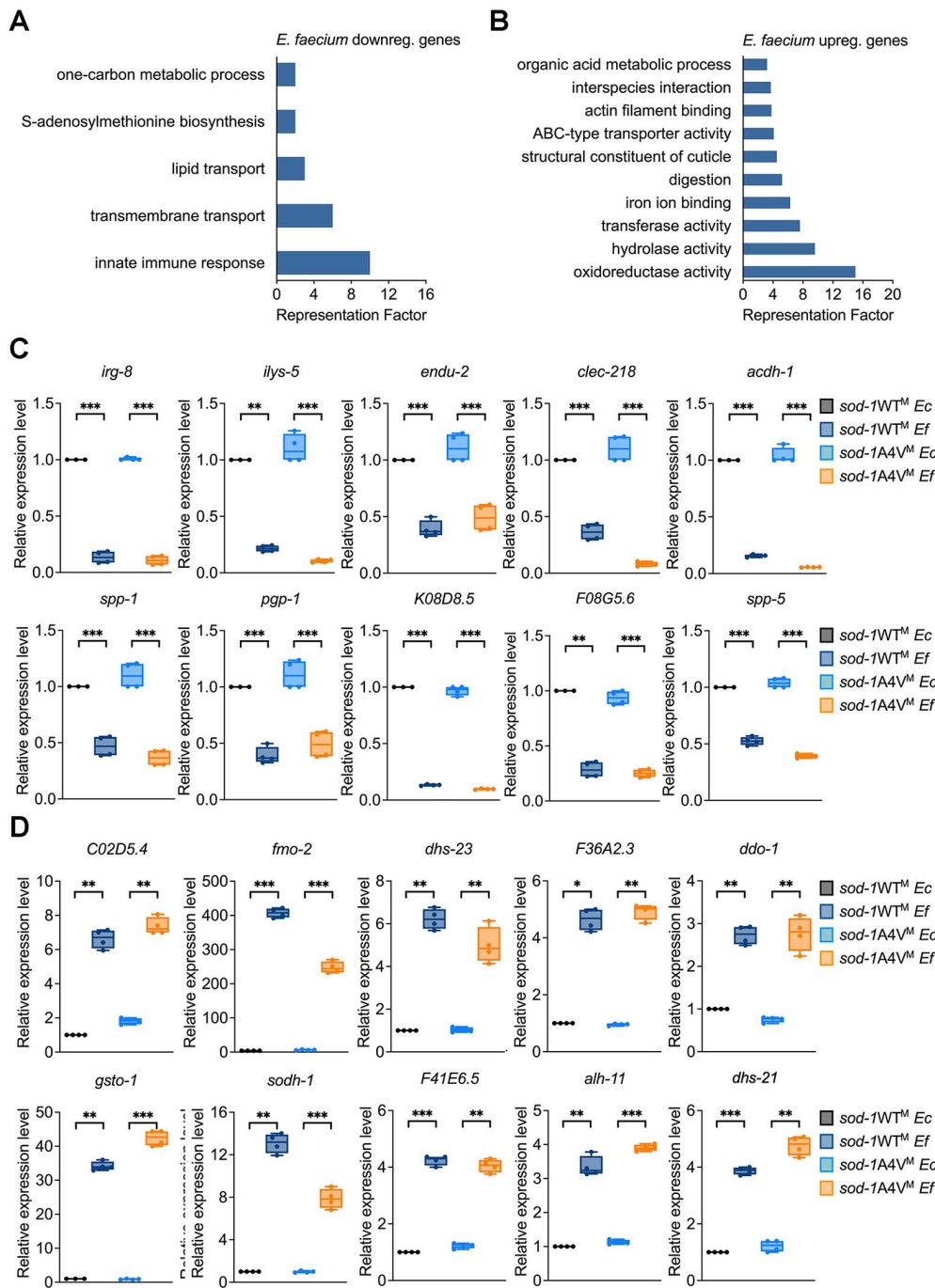

**Fig 2. Gene signature upon *Enterococcus faecium* exposure. (A)** GO analysis of (A) down-regulated and (B) up-regulated genes in *sod-1* WT^M and *sod-1* A4V^M animals exposed to *Escherichia coli* or *E. faecium*. **(B)** qRT-PCR validation of the innate immune response genes that were down-regulated with *E. faecium* treatment (*N* = 3 biological replicates). **(C)** qRT-PCR validation of the oxidoreductase activity genes that were up-regulated with *E. faecium* treatment (*N* = 3 biological replicates). Values are expressed as the fold difference compared with *sod-1* WT^M animals fed on *E. coli* ± SD by one-way ANOVA with Tukey's multiple comparisons test, * *P* < 0.05, ** *P* < 0.01, *** *P* < 0.001. The data underlying this Figure can be found in S1 Data.

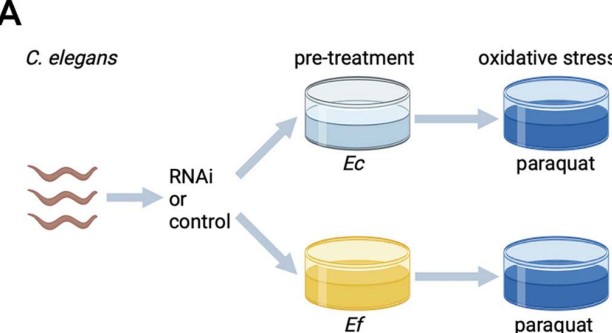

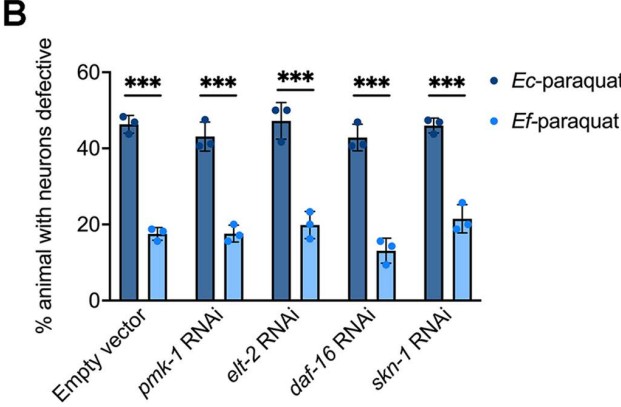

**Fig 3. *Enterococcus faecium*-mediated neuroprotection in *Caenorhabditis elegans* does not require classic immune pathways. (A)** Schematic of RNAi-specific genes, pretreatment with probiotic *E. faecium*, and paraquat exposure. Gravid *sod-1* A4V^M animals laid eggs on RNAi and control plates. The eggs hatched and grew into young adults, which were then treated on *E. faecium* or *Escherichia coli* lawns for 24 hours before a 24-hour paraquat exposure. Figure created using BioRender. Sang, Y. (2026) https://BioRender.com/fg7i4lx. **(B)** Oxidative stress induced cholinergic motor neuron degeneration in *sod-1* A4V^M animals. *sod-1* A4V^M animals were fed control or *pmk-1, elt-2, daf-16,* and *skn-1* RNAi. After growing to young adults on RNAi plates, the animals were treated with *E. faecium* or *E. coli* for 24 hours before a 24-hour paraquat exposure. Cholinergic motor neuron integrity was accessed by scoring for the absence of at least two cholinergic motor neurons posterior to the vulva. * $P < 0.05$, ** $P < 0.01$, *** $P < 0.001$. RNAi, RNA interference. The data underlying this Figure can be found in S1 Data.

## Cytochrome P450 genes protect against motor neuron degeneration during oxidative stress

Our transcriptomic analysis of *E. faecium*-treated animals revealed a strong enrichment of oxidoreductase genes (Fig 2B), suggesting that these genes may play a role in neuroprotection. Notably, 10 out of the 23 genes in the oxidoreductase activity cluster belong to the cytochrome P450 (CYP) family (S2 Table), a group previously linked to neurodegenerative processes [17,38]. qRT-PCR analysis confirmed that multiple *cyp* genes, including *cyp-35A3, cyp-35A1, cyp-35C1, cpy-33C8, cpy-34A9, cyp-35D1, cpy-13A7, cpy-33C9, cyp-35A5,* and *cyp-13A5*, were up-regulated upon *E. faecium* exposure in *sod-1* WT^M and *sod-1* A4V^M animals (S7 Fig).

Some CYP enzymes contribute to ROS generation [39,40], while others help defense against oxidative stress [41–43]. To test whether CYPs are required in *E. faecium*-mediated neuroprotection, we knocked down several *cyp* genes up-regulated by *E. faecium*. Due to sequence similarity and redundancy, *cyp-35A12345* can be knocked down by mixed RNAi, which targets all five genes [44]. Knock down of *cyp-35A12345* decreased *E. faecium*-mediated protection against motor neuron degeneration (Figs 4A and S8). We also detected the survival of animals and locomotion with *cyp-35A12345* knockdown and found that inhibition of *cyp-35A12345* also decreased animal survival (Fig 4B) and locomotion (Fig 4C) protection conferred by *E. faecium* exposure.

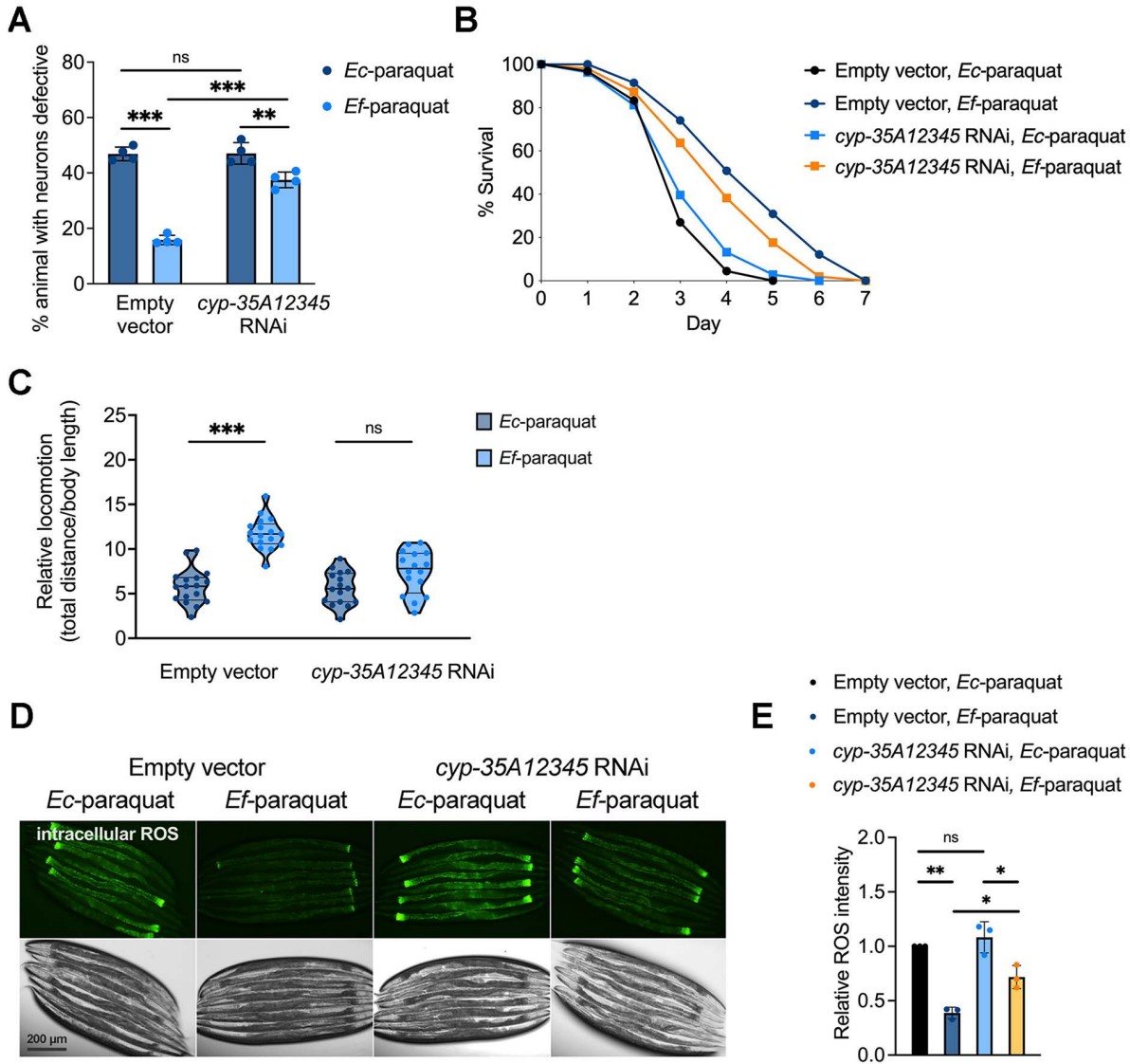

**Fig 4. Cytochrome P450s are required in *Enterococcus faecium*-mediated neuroprotection. (A)** Oxidative stress induced cholinergic motor neuron degeneration in *sod-1* A4V^M animals. *sod-1* A4V^M animals were fed HT115 expressing control or *cyp-35A12345* RNAi. After growing to young adults on RNAi plates, the animals were treated with *E. faecium* or *Escherichia coli* for 24 hours before a 24-hour paraquat exposure. Cholinergic motor neuron integrity was accessed by scoring for the absence of at least two cholinergic motor neurons posterior to the vulva. * $P < 0.05$, ** $P < 0.01$, *** $P < 0.001$. **(B)** Survival curve of *cyp-35A12345* knockdown with *E. faecium* pretreatment, followed by paraquat-induced oxidative stress. *sod-1* A4V^M animals were fed HT115 expressing control or *cyp-35A12345* RNAi and grown into young adults. After 24 hours of paraquat exposure, the survival rate of these animals, pretreated on *E. faecium* and *E. coli* lawns, was recorded. Animals were transferred to new paraquat plates daily until all animals stopped laying eggs. **(C)** Locomotion analysis of *sod-1* A4V^M animals after *cyp-35A12345* knockdown, following *E. faecium* pretreatment and paraquat exposure. *sod-1* A4V^M animals were fed HT115 expressing control or *cyp-35A12345* RNAi and grown into young adults. Each young adult was pretreated on *E. faecium* or *E. coli* lawn, then exposed to 5mM paraquat for 48 hours. The ratio of movement distance to body length, measured using NIH ImageJ software, was used as a movement index. **(D)** Intracellular ROS analysis of *sod-1* A4V^M animals after *cyp-35A12345* knockdown, following *E. faecium* pretreatment and paraquat exposure. Intracellular ROS levels were shown in green. **(E)** Quantification of DCF fluorescence intensity in (D), $n = 25$. The "n" represents the number of animals in each experiment. $N = 3$ biological replicates. The data underlying this Figure can be found in S1 Data.

To explore the mechanism of CYP-mediated neuroprotection during oxidative stress, we measured intracellular ROS levels using the fluorescent probes 2′,7′-dichlorofluorescein diacetate (H2DCFDA) [45]. Animals fed on *E. coli* followed by paraquat treatment showed high ROS levels, while *E. faecium*-treated animals showed low ROS levels when exposed to paraquat. Knocking down *cyp-35A12345* decreased the reduction of ROS levels conferred by *E. faecium* exposure (Fig 4D and 4E). Together, these results demonstrate that CYPs are required in *E. faecium*-mediated neuroprotection and function, at least in part, by reducing intracellular ROS levels during oxidative stress.

## NHR-86 is required in *E. faecium*-mediated neuroprotection

CYP enzymes are central to xenobiotic metabolism and are tightly regulated by NHRs in mammals and *C. elegans* [46–49]. In response to xenobiotic exposure, NHRs function as transcription factors that activate genes encoding metabolic enzymes and components of multi-drug efflux pumps [50]. To determine whether NHRs mediate *E. faecium*-induced expression of *cyp* genes, we first identified candidate *nhr* genes involved in this regulatory pathway.

A previous transcriptomic study found that both *cyp* and *nhr* genes were up-regulated in *C. elegans* exposed to genotoxic compounds, suggesting coordinated regulation of these gene families [48,51]. We performed a candidate mutant screen targeting 13 *nhr* genes responsible for regulating *cyp* genes (*nhr-62, nhr-130, nhr-201, nhr-203, nhr-237, nhr-86, nhr-207, nhr-235, nhr-11, nhr-12, nhr-106, nhr-196,* and *nhr-205*) using *cyp-35A1, cyp-35A3,* and *cyp-35A5* as transcriptional readouts. Treatment with *E. faecium* in wild-type animals led to robust induction of *cyp-35A1* expression. This induction was significantly reduced by RNAi targeting either *nhr-203* or *nhr-86* (Fig 5A), indicating that both nuclear receptors contribute to the regulation of *cyp-35A1*. In contrast, the expression of *cyp-35A3* and *cyp-35A5* in response to *E. faecium* depended solely on *nhr-86*, as their mRNA levels were unaffected by *nhr-203* knockdown (S9 Fig).

We next tested whether *nhr-86* or *nhr-203* was required on *E. faecium*-mediated neuroprotection. Knocking down *nhr-86*, but not *nhr-203*, impaired *E. faecium*-mediated protection against motor neuron degeneration (Figs 5B and S10) and reduced the suppression of ROS levels (Figs 5C and S11). These findings were further supported by large particle flow cytometry, which showed that *E. faecium*-induced reductions in intracellular ROS were dependent on *nhr-86* (Fig 5D and 5E), but not on *nhr-203* (S12 Fig). Additionally, *nhr-86* knockdown abolished the protective effects of *E. faecium* on oxidative stress-induced locomotion and survival (Fig 5F and 5G). Together, these data indicate that *nhr-86*, but not *nhr-203*, is essential for *E. faecium*-mediated neuroprotection and functions upstream of CYP gene activation, while *nhr-86* expression itself is not regulated by *E. faecium*, as shown by RNA-seq (S1 Table) and qRT-PCR (S13 Fig).

To further assess the role of NHR-86 in the transcriptional program activated by *E. faecium*, we compared the *E. faecium*–regulated transcriptome with the published NHR-86 ChIP-seq dataset [52]. This analysis revealed an overlap between *E. faecium*-induced genes and NHR-86-bound targets, including CYP genes such as *cyp-35A1* and *cyp-35A5*, as well as numerous other genes (S1 Table). These findings indicate that NHR-86 acts as one of several transcriptional effectors mediating the host response to *E. faecium*.

To confirm that *nhr-86* is required for this protective pathway (Fig 5B–5G), we tested whether expression of *nhr-86* under its endogenous promoter could rescue the reduced neuroprotection in *sod-1* A4V$^M$ animals. These animals were generated by crossing *nhr-86(tm2590);nhr-86p::nhr-86* animals with *sod-1* A4V$^M$ animals (Fig 6A). Expression of NHR-86 fully restored *E. faecium*-mediated protection against cholinergic motor neuron degeneration in *sod-1* A4V$^M$ animals (Fig 6B), and rescued the mRNA levels of *cyp-35A1, cyp-35A3,* and *cyp-35A5* upon *E. faecium* exposure in *sod-1* A4V$^M$;*nhr-86(tm2590)* animals (Figs 6C and S14), and restored the protective effects of *E. faecium* on animal survival (Fig 6D–6F) and locomotion (Fig 6G). These results indicate that the expression of *nhr-86* under its own promoter is sufficient to reestablish CYP activation and confer neuroprotection in response to *E. faecium.* Notably, overexpression of NHR-86 alone did not confer neuroprotection or induce *cyp* expression on an *E. coli* diet (Fig 6B and 6C), indicating that NHR-86 activation depends on specific signals provided by *E. faecium*.

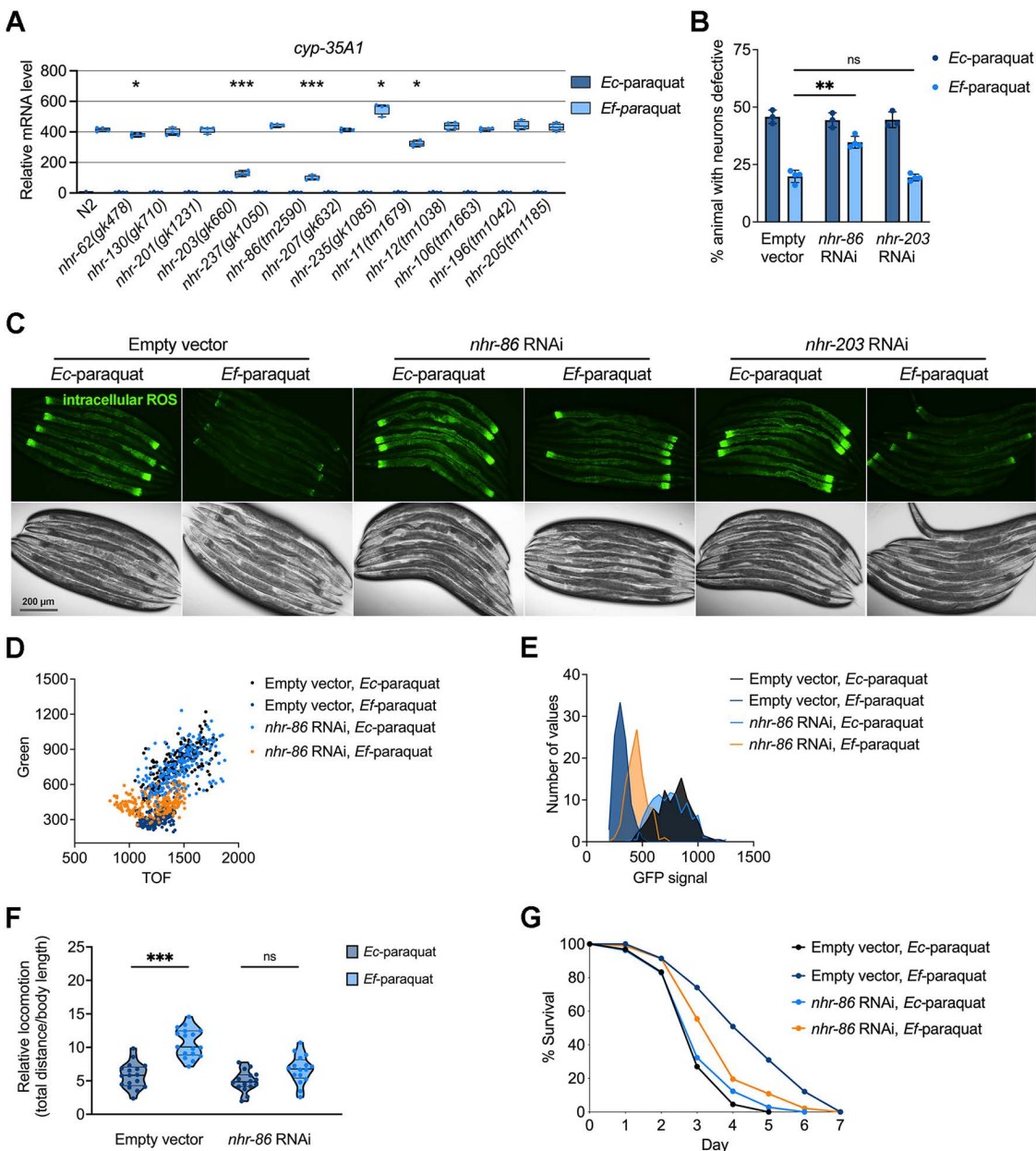

**Fig 5. NHR-86 is required for *Enterococcus faecium*-mediated neuroprotection. (A)** Screening of *nhr* genes using *cyp-35A1* as a reporter. qRT-PCR analysis of *cyp-35A1* mRNA levels in *nhr* mutants. Young adult *nhr* mutant animals pretreated on *E. faecium* or *Escherichia coli* lawns were harvested for RNA extraction and qRT-PCR assay. (*N* = 3 biological replicates). Values represent fold differences relative to animals fed on *E. coli*, analyzed by one-way ANOVA with Tukey's multiple comparisons test, * *P* < 0.05, ** *P* < 0.01, *** *P* < 0.001. **(B)** Oxidative stress induced cholinergic motor neuron degeneration in *sod-1* A4V$^M$ animals. *sod-1* A4V$^M$ animals were fed HT115 expressing *nhr-86*, *nhr-203* or empty vector as a control. After growing to young adults on RNAi plates, the animals were treated with *E. faecium* or *E. coli* for 24 hours before a 24-hour paraquat exposure. Cholinergic motor neuron integrity was accessed by scoring for the absence of at least two cholinergic motor neurons posterior to the vulva. * *P* < 0.05, ** *P* < 0.01, *** *P* < 0.001. RNAi, RNA interference. **(C)** Intracellular ROS analysis of *sod-1* A4V$^M$ animals after *nhr-86* or *nhr-203* knockdown, following *E. faecium* pretreatment and paraquat exposure. Intracellular ROS levels were shown in green. **(D)** Dot-plot representation of green fluorescence intensity vs. TOF of vector or *nhr-86* RNAi animals exposed to *E. faecium* or *E. coli* followed by paraquat exposure (*n* = 300). **(E)** Frequency distribution of green fluorescence in vector or *nhr-86* RNAi animals exposed to *E. faecium* or *E. coli* followed by paraquat exposure. Three independent experiments were conducted. Empty vector*, Ec*-paraquat vs. Empty vector*, Ef*-paraquat, *P* < 0.0001; Empty vector*, Ec*-paraquat vs. *nhr-86* RNAi*, Ec*-paraquat, *P* = 0.1240; Empty vector*, Ef*-paraquat vs. *nhr-86* RNAi*,Ef*-paraquat, *P* < 0.0001; *nhr-86* RNAi*, Ec*-paraquat vs. *nhr-86* RNAi*, Ef*-paraquat, *P* < 0.0001. **(F)** Locomotion analysis of *sod-1* A4V$^M$ animals after *nhr-86* knockdown, following *E. faecium* pretreatment and paraquat exposure. *sod-1* A4V$^M$ animals were fed

HT115 expressing *nhr-86* and grown to young adults. Each young adult was pretreated on *E. faecium* or *E. coli* lawn, then exposed to 5mM paraquat for 48 hours. The ratio of movement distance to body length, measured using NIH ImageJ software, was used as a movement index. **(G)** Survival curve of *nhr-86* knockdown with *E. faecium* pretreatment, followed by paraquat-induced oxidative stress. *sod-1* A4V^M animals were fed HT115 expressing *nhr-86* and grown to young adults. After 24 hours of paraquat exposure, the survival rate of these animals, pretreated on *E. faecium* and *E. coli* lawns, was recorded. Animals were transferred to new paraquat plates daily until all animals stopped laying eggs, and survival was scored. The data underlying this Figure can be found in S1 Data.

## The intestinal function of NHR-86 is required in *E. faecium*-mediated neuroprotection

NHR-86 is expressed in neurons, the intestine, and hypodermis [53]; however, because the CYP genes it regulates are primarily intestinal [54] and neurodegeneration occurs in neurons, we focused on its neuronal and intestinal roles.

To determine whether the neuronal NHR-86 function is required for neuroprotection during oxidative stress, we performed a neuron-specific knockdown of *nhr-86* in *sod-1* A4V^M animals. We generated a neuron-specific RNAi ALS model by crossing *sod-1* A4V^M animals with the neuron-specific RNAi strain MAH677. Neuron-specific knockdown of *nhr-86* had no effect on *E. faecium-mediated* neuroprotection (S15A and S15B Fig). *dnc-1* served as a positive control, as its knockdown causes cholinergic motor neuron degeneration [55]. *smn-1* was used as a negative control, as its neuron-specific RNAi affects only GABAergic motor neurons but does not impact cholinergic motor neurons [56]. Together, these results indicate that the neuronal function of NHR-86 is not required for the neuroprotective effect of *E. faecium*.

To determine whether the intestinal function of NHR-86 is required for neuroprotection during oxidative stress, we expressed NHR-86 specifically in the intestine of *sod-1* A4V^M animals (Fig 7A). For intestine-specific expression, *nhr-86* was placed under the control of *vha-6* promoter, which is active in intestinal cells [57]. Intestine-specific expression of NHR-86 fully restored *E. faecium*-mediated protection against motor neuron degeneration in *sod-1* A4V^M animals (Fig 7B), and rescued the mRNA levels of *cyp-35A1*, *cyp-35A3*, and *cyp-35A5* in *sod-1* A4V^M;*nhr-86(tm2590)* animals (Figs 7C and S16). It also restored the protective effect of *E. faecium* on animal survival (Figs 6D, 6E, and 7D) and locomotion (Fig 7E). Together, these results indicate that the intestinal expression of NHR-86 is sufficient to reestablish CYP activation and confer neuroprotection upon exposure to *E. faecium.*

## Discussion

Our study demonstrates that probiotic *E. faecium* protects against oxidative stress-induced neurodegeneration in *C. elegans* (Fig 8), consistent with previous findings in mice showing antioxidant and neuromodulatory properties of *E. faecium* [58]. Rather than acting through classic immune pathways such as PMK-1/p38 MAPK [28–30], DAF-16/FOXO [31–33], ELT-2/GATA [34,35], and SKN-1/Nrf2 [36,37], which are down-regulated at the transcriptional level but are not required for protection in our RNAi assays, *E. faecium*-mediated neuroprotection depends on the transcription factor NHR-86. We show that NHR-86 is required for the transcriptional activation of a subset of CYP gene products that mitigate ROS and preserve neuronal function. Among the 284 NHRs encoded by the *C. elegans* genome [59,60], NHR-86 is the first linked to protection against neurodegeneration. Our work indicates that NHR-86 is a key regulator of probiotic-mediated stress resistance and reveals a functional NHR-CYP axis essential for preserving motor neuron integrity under oxidative conditions.

In addition to the NHR-86–CYP axis described here, several intestinal pathways in *C. elegans* have been implicated in neuronal protection, underscoring the importance of gut–neuron communication. For example, a mitochondria-regulated immune pathway in the intestine confers neuroprotection by coordinating stress responses across tissues [61]. Intestinal signaling through the *rab-27* pathway regulates axon regeneration, highlighting a direct role for gut-derived signals in neuronal repair [62]. Additionally, metabolic intestine-to-neuron signaling, such as propionate supplementation, rescues α-synuclein-induced neurodegeneration [20], while gut-derived Wnt endocrine signaling regulates synaptic assembly in the nervous system [63]. Together, these findings establish that intestinal pathways—including mitochondrial stress responses, neurotransmitter signaling, metabolic circuits, and endocrine communication—contribute to neuronal

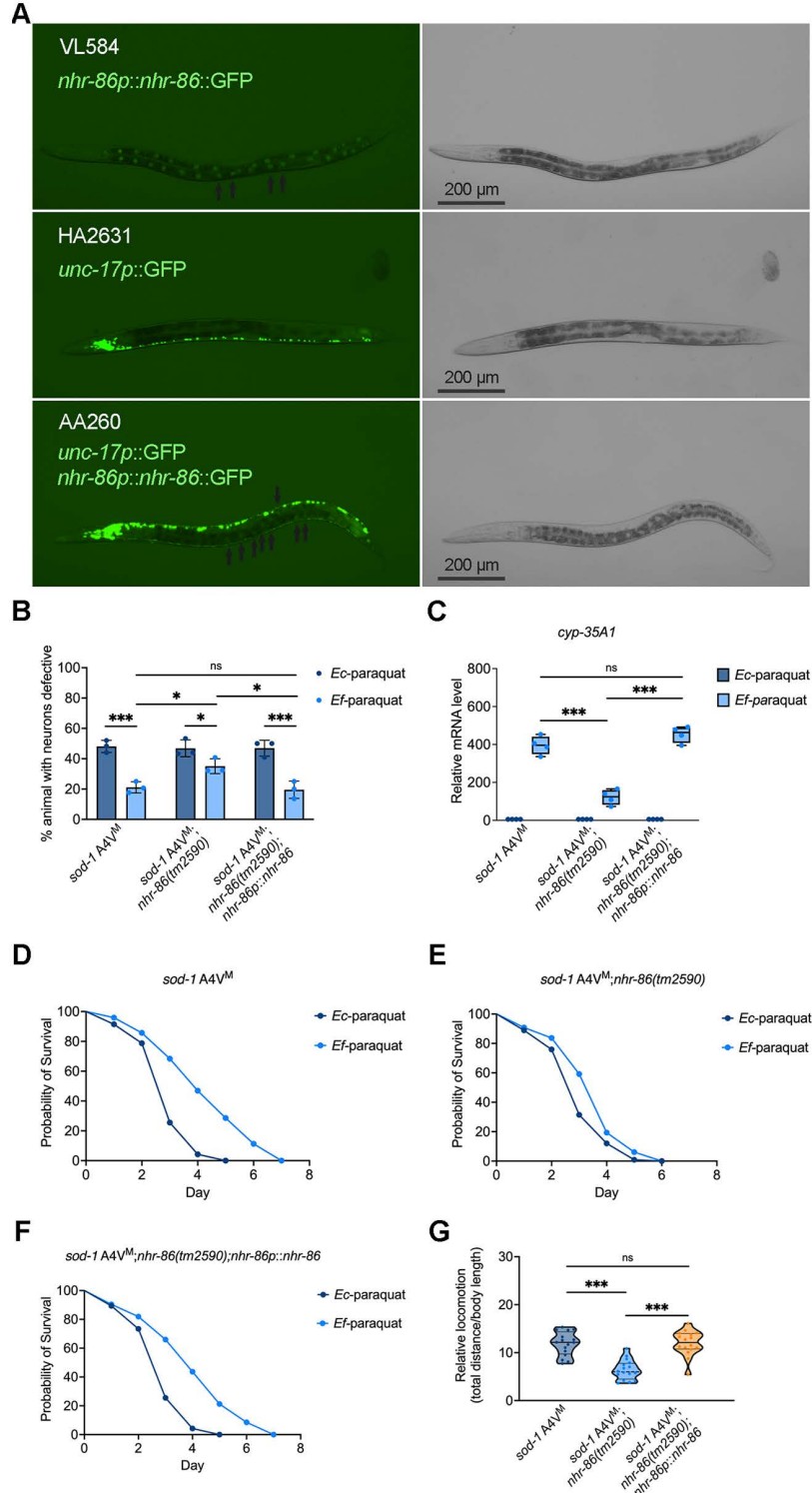

**Fig 6. Native nhr-86 expression rescues *sod-1* A4V^M;*nhr-86(tm2590)* animals under oxidative stress. (A)** GFP fluorescence and corresponding brightfield images showing expression patterns of transgenes in different animals. Strain VL584 (*nhr-86(tm2590);nhr-86*p::*nhr-86*::*gfp*) expresses *nhr-86p*::GFP (arrows). Strain HA2631 (*sod-1* A4V^M) expresses *unc-17p*::GFP in cholinergic motor neurons. Strain AA260 co-expresses *unc-17p*::GFP and *nhr-86p*::GFP, highlighting both neuronal and intestinal expression domains. Scale bar: 200 μm. **(B)** Oxidative stress induced cholinergic motor

neuron degeneration in *sod-1* A4V^M, *sod-1* A4V^M;*nhr-86(tm2590)*, and *sod-1* A4V^M;*nhr-86(tm2590)*;*nhr-86p*::*nhr-86* animals treated with *Enterococcus faecium* or *Escherichia coli* for 24 hours before a 24-hour paraquat exposure. Cholinergic motor neuron integrity was assessed by scoring for the absence of at least two cholinergic motor neurons posterior to the vulva. * *P* < 0.05, ** *P* < 0.01, *** *P* < 0.001. **(C)** mRNA level of *cyp-35A1* in *sod-1* A4V^M, *sod-1* A4V^M;*nhr-86(tm2590)*, and *sod-1* A4V^M;*nhr-86(tm2590)*;*nhr-86p*::*nhr-86* animals. (*N* = 3 biological replicates). Values are expressed as the fold difference compared with worms fed on *E. coli* ± SD by one-way ANOVA with Tukey's multiple comparisons test, * *P* < 0.05, ** *P* < 0.01, *** *P* < 0.001. **(D)** Survival curve of *sod-1* A4V^M animals with *E. faecium* or *E. coli* pretreatment, followed by paraquat-induced oxidative stress. **(E)** Survival curve of *sod-1* A4V^M;*nhr-86(tm2590)* animals with *E. faecium* or *E. coli* pretreatment, followed by paraquat-induced oxidative stress. **(F)** Survival curve of *sod-1* A4V^M;*nhr-86(tm2590)*;*nhr-86p*::*nhr-86*::*gfp* animals with *E. faecium* and *E. coli* pretreatment, followed by paraquat-induced oxidative stress. **(G)** Locomotion analysis of *sod-1* A4V^M, *sod-1* A4V^M;*nhr-86(tm2590)*, and *sod-1* A4V^M;*nhr-86(tm2590)*;*nhr-86p*::*nhr-86* animals following *E. faecium* pretreatment and paraquat exposure. The data underlying this Figure can be found in S1 Data.

resilience. Our identification of NHR-86 as a mediator of probiotic-induced neuroprotection places the NHR-86–CYP axis as a novel intestinal mechanism that complements previously described gut-driven protective pathways.

Our results establish NHR-86 as a central mediator of *E. faecium*-induced neuroprotection, acting upstream of cytochrome P450 genes to regulate oxidative stress responses. This regulatory relationship mirrors the function of HNF4 in human hepatocytes, where it controls the expression of multiple CPYs [64]. Although HNF4 has not previously been linked to neurodegenerative disorders, our findings raise the possibility that similar NHR-CYP regulatory circuits may operate in humans and influence neuronal survival under oxidative stress. This provides guidance to explore more functions of the key factor HNF4 in humans.

In our model, the intestinal probiotic *E. faecium* protects motor neurons during oxidative stress induced by paraquat. Mechanistically, *E. faecium* induces the expression of CYP enzymes, and the intestinal function of the nuclear hormone receptor NHR-86 is required for CYP induction and neuroprotection. These CYPs reduce intracellular ROS generated by paraquat-induced oxidative stress. Because ROS are diffusible, lowering their ROS levels in the intestine also reduces oxidative stress in neighboring motor neurons. Thus, the probiotic indirectly protects neurons by enhancing intestinal antioxidant capacity. These findings provide direct experimental evidence that intestinal responses to microbial signals can modulate neuronal resilience under environmental oxidative stress.

*E. faecium*-mediated neuroprotection is observed only in mutants with heightened stress sensitivity, such as *sod-1* A4V^M, but not in *sod-1* WT^M animals, where baseline stress is insufficient to induce degeneration. These findings indicate that ALS-linked SOD-1 mutations prime neurons for stress-induced degeneration, providing a context in which probiotics can exert measurable protective effects and highlighting the relevance of the *sod-1* A4V^M model for studying ALS-related neurodegeneration. While we focused on *sod-1* A4V^M-mediated motor neuron degeneration, our results in the *sod-1* G85R^M model indicate that *E. faecium* can also protect glutamatergic neurons, suggesting that its neuroprotective effects may extend to other ALS-associated mutations; future studies could explore its impact in additional ALS or Frontotemporal Dementia models.

Our data also strengthen the case for CYP enzymes as core effectors of cellular defense against oxidative stress. In diverse species, CYPs contribute to detoxification by metabolizing xenobiotics and reactive metabolic intermediates. In *Daphnia magna*, CYP360A8 mitigates paraquat-induced toxicity [65], and in *C. elegans*, CYP-35A2 and CYP-35B1 function in xenobiotic detoxification and stress adaptation [66]. Several mammalian CYPs also promote neuronal survival by eliminating toxic byproducts of cellular metabolism [67]. Beyond detoxification, CYPs metabolize polyunsaturated fatty acids into epoxyeicosatrienoic acids, which activate protective pathways that buffer free radicals and prevent subcellular damage [43]. Additionally, CYP152 enzymes act as peroxygenases that utilize hydrogen peroxide to catalyze fatty acid oxidation, facilitating efficient metabolism while maintaining cellular redox balance and mitigating oxidative stress [68]. Our identification of *cyp* genes as essential mediators of *E. faecium*-induced neuroprotection underscores their functional relevance in stress resilience.

These findings add to the growing body of evidence linking gut microbial signals to neuronal health. The gut–brain axis, increasingly recognized as a key modulator of neurodegenerative disease, is influenced by microbiota-derived metabolites that shape central nervous system function [69]. Microbial-derived short-chain fatty acids, generated during bacterial fermentation, possess neuroprotective properties and impact neuronal health and behavioral responsiveness [70]. Additionally,

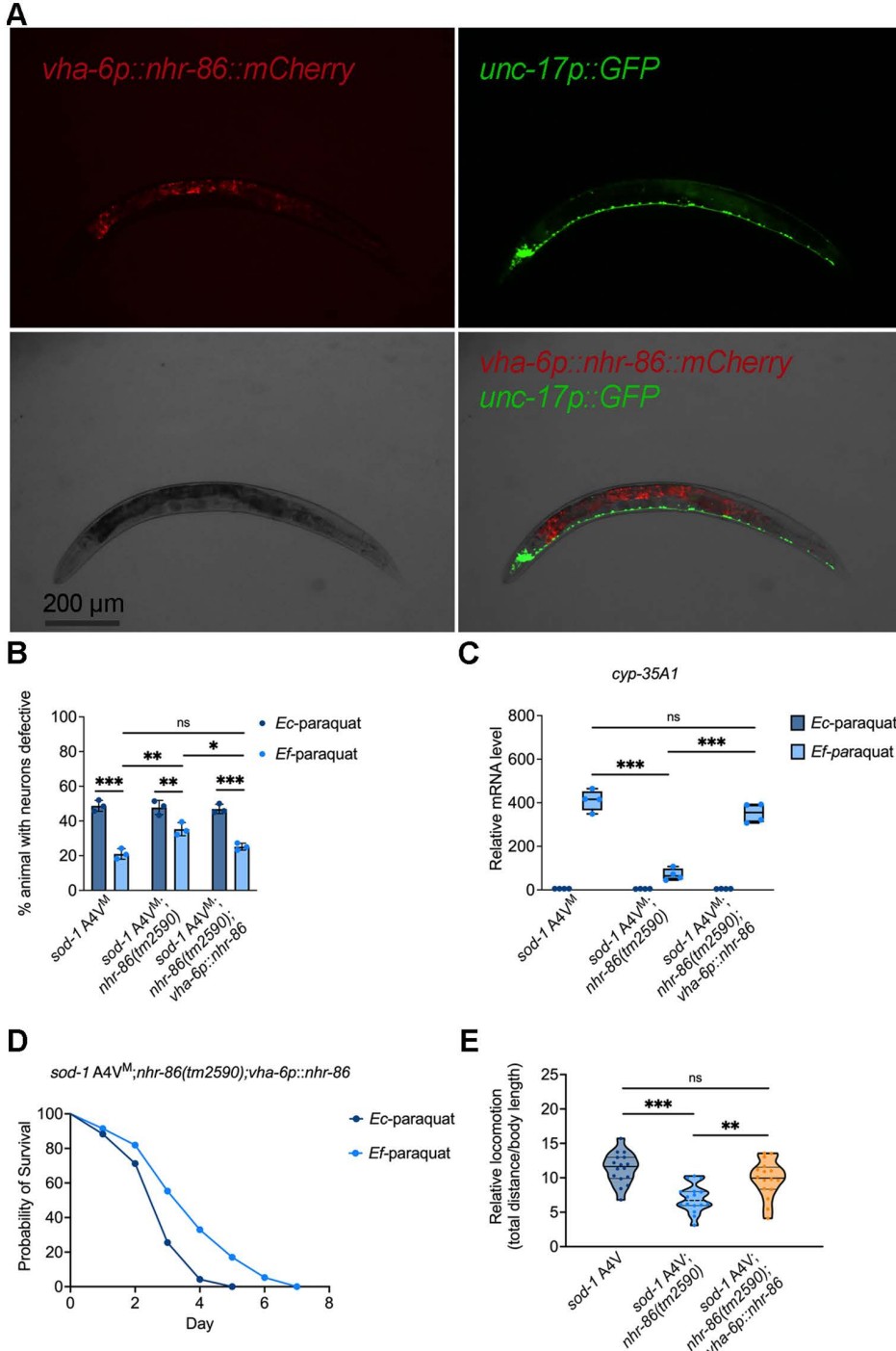

**Fig 7. Intestinal *nhr-86* expression rescues *sod-1* A4V^M;*nhr-86(tm2590)* animals under oxidative stress. (A)** Representative fluorescence micrographs of *sod-1* A4V^M;*nhr-86(tm2590)*;*nhr-86p*::*nhr-86*::*mCherry* animals. *n* = 30; representative of four independent experiments. Scale bars indicate 200 μm. **(B)** Oxidative stress induced cholinergic motor neuron degeneration in *sod-1* A4V^M, *sod-1* A4V^M;*nhr-86(tm2590)*, and *sod-1* A4V^M;*nhr-86 (tm2590)*;*vha-6p*::*nhr-86* animals treated with *Enterococcus faecium* or *Escherichia coli* for 24 hours before a 24-hour paraquat exposure. **(C)** mRNA level of *cyp-35A1* in *sod-1* A4V^M, *sod-1* A4V^M;*nhr-86(tm2590)*, and *sod-1* A4V^M;*nhr-86(tm2590)*;*vha-6p*::*nhr-86* animals. (*N* = 3 biological replicates). Values are expressed as the fold difference compared with worms fed on *E. coli* ± SD by one-way ANOVA with Tukey's multiple comparisons test, *** $P < 0.001$. **(D)** Survival curve of *sod-1* A4V^M;*nhr-86(tm2590)*; *vha-6p*::*nhr-86*::*gfp* with *E. faecium* or *E. coli* pretreatment, followed by paraquat-induced oxidative stress. **(E)** Locomotion analysis of *sod-1* A4V^M, *sod-1* A4V^M;*nhr-86(tm2590)*, and *sod-1* A4V^M;*nhr-86(tm2590)*;*vha-6p*::*nhr-86* animals following *E. faecium* pretreatment and paraquat exposure. The data underlying this Figure can be found in S1 Data.

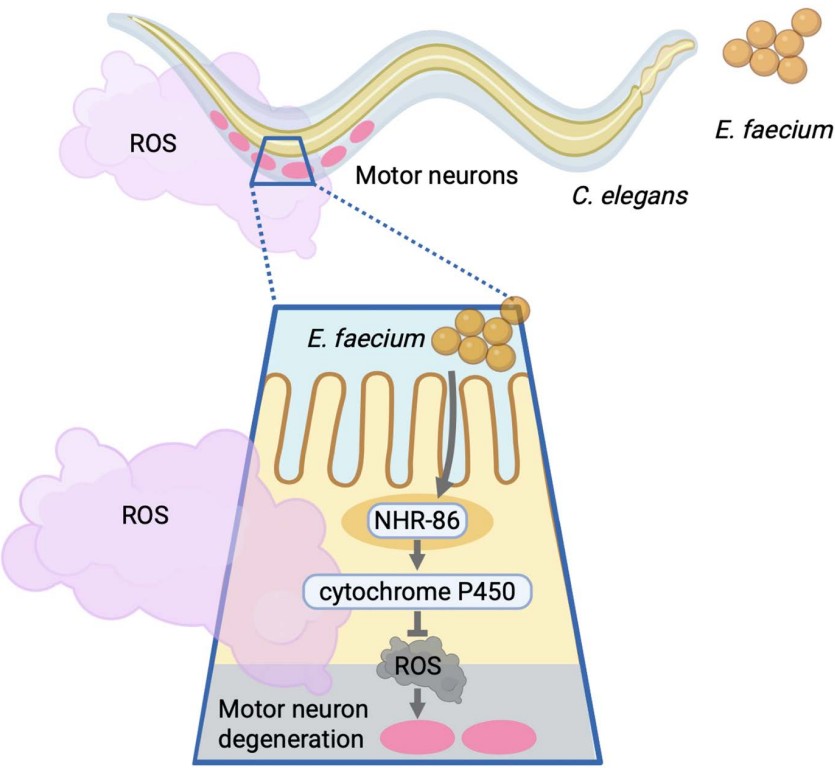

**Fig 8. Model of *Enterococcus faecium*-mediated neuroprotection under oxidative stress.** The intestinal probiotic *E. faecium* protects motor neurons from degeneration during paraquat-induced oxidative stress by inducing cytochrome P450 (CYP). This induction requires the nuclear hormone receptor NHR-86. CYP enzymes detoxify reactive oxygen species (ROS) generated during oxidative stress, reducing ROS burden and protecting motor neurons. This model highlights an NHR-CYP regulatory axis through which gut microbial signals enhance neuronal resilience under environmental stress. Figure created using BioRender. Sang, Y. (2026) https://BioRender.com/fg7i4lx.

tryptophan from gut microbiota is essential for serotonin production, and disruptions in neurotransmitter balance can lead to mood disorders, cognitive decline, and neurodegeneration [71,72]. Our studies indicate that *E. faecium* is an intestinal probiotic capable of triggering a host transcriptional response with systemic protective outcomes. By activating NHR-86 in the intestine and inducing CYP expression, *E. faecium* promotes resistance to oxidative stress in motor neurons without engaging classical immune pathways, distinguishing this mechanism from those described for other lactic acid bacteria.

Together, our study defines a gut-driven, transcription factor-dependent mechanism by which a probiotic bacterium protects neurons from oxidative stress-induced degeneration. This work provides new insight into how microbial interactions at the intestinal interface can regulate host gene expression to prevent neurodegeneration. While further studies are needed to assess the relevance of this mechanism in higher organisms, our findings highlight a conserved transcriptional program that may be useful to mitigate neurodegeneration.

## Materials and methods

### Bacterial strains

The following bacterial strains were used: *Escherichia coli* OP50, *E. coli* HT115(DE3), *Enterococcus faecium* NCTC 7171. *E. faecium* was grown in brain–heart infusion (BHI) medium at 37 °C, and the cultures of other bacteria were grown in Luria-Bertani (LB) broth at 37 °C.

### C. elegans strains

All *C. elegans* strains were maintained on nematode growth medium (NGM) and fed *E. coli* strain OP50. The *C. elegans* strains wild-type N2 Bristol, *nhr-62(gk478)*, *nhr-86(tm2590)*, *nhr-130(gk710)*, *nhr-235(gk1085)*, *nhr-178(gk1005)*, *nhr-178(gk1158)*, *nhr-201(gk1231)*, *nhr-203(gk660)*, *nhr-207(gk632)* and *nhr-86(tm2590);nhr-86p::nhr-86(ORF)::GFP* used were obtained from the *Caenorhabditis* Genetics Center (University of Minnesota, Minneapolis, MN). *nhr-142(gk826)*, *nhr-237(gk1050)*, *nhr-11(tm1679)*, *nhr-12(tm1038)*, *nhr-106(tm1663)*, *nhr-176(tm954)*, *nhr-196(tm1042)*, and *nhr-205(tm1185)* animals were obtained from National Bioresource Project (NBRP), Japan. HA2630 *sod-1* WT$^M$ and HA2631 *sod-1* A4V$^M$ strain were gifts from Dr. Anne Hart [22]. Detailed strain information is listed in S3 Table.

### Probiotic *E. faecium* pretreatment and paraquat exposure

Synchronized young adult animals were washed three times in M9 buffer and transferred to *E. faecium* lawns grown on 2% agar BHI plates for colonization for 1 day at 25 °C. The worms were then washed and transferred to paraquat agar plates containing OP50 lawn. A 250 mM stock solution of paraquat (Sigma-Aldrich 856177) was prepared in water and added to NGM media to achieve the 5 mM paraquat concentration. *E. coli* OP50 lawns were used as pretreatment control.

### Fluorescent microscopy and cholinergic motor neurons integrity assessment

To assess the integrity of cholinergic motor neurons in *C. elegans* following paraquat exposure, we exposed HA2630 or HA2631 animals expressing *unc-17p*::GFP to 5 mM paraquat according to the specified protocol. After 24 hours, the animals were washed three times with M9 buffer, immobilized using 30 mM sodium azide (Sigma) in M9 buffer, mounted on a 2% agarose pad, and visualized using a Leica M165 FC fluorescence stereomicroscope. Defective cholinergic motor neurons were identified by scoring for the absence of at least two cholinergic motor neurons posterior to the vulva. All image acquisition and manual scoring were performed with researchers blinded to genotype and treatment.

### Survival assay

Following 24 hours of paraquat exposure, the survival of *sod-1* A4V$^M$ animals pretreated on *E. faecium* or *E. coli* lawns was recorded. Time point 0 was defined as the completion of the 24-hour paraquat treatment. Animals were transferred to new paraquat plates daily until all animals stopped laying eggs, and survival was scored.

### Locomotion phenotype analysis

An image-based assay was used to assess the locomotion speed of *C. elegans*. Animals were transferred to a plate with a fresh bacterial lawn to trace movement tracks. Upon release, worms exhibited a maximum movement response for a short duration. A 15-s movie was recorded for each worm, and a picture was taken at the 15th second. The ratio of movement distance to body length, measured using NIH ImageJ software, was used as a movement index [73]. Measurements were taken 48 hours after transferring to 5 mM paraquat.

### Glutamatergic neuron degeneration

Day 1 adult animals, pretreated with *E. faecium* and subsequently exposed to paraquat, were washed off the plates with M9 buffer and incubated with DiD (Fisher DiIC18(5), D307) in a microcentrifuge tube. After 1 hour, animals were transferred to a regular NGM plate. After another hour, animals were immobilized with 20 mM sodium azide in M9 buffer, mounted on a 2% agar pad, and covered with a glass coverslip. Fluorescent neuronal cell bodies were visualized and scored for lack of dye uptake under a Leica M165 FC fluorescence microscope.

## RNA interference

RNA interference (RNAi) was used to generate loss-of-function phenotypes by feeding nematodes with *E. coli* strain HT115(DE3) expressing double-stranded RNA homologous to a target gene [74,75]. *E. coli* with the appropriate vectors were grown overnight at 37 °C in LB broth containing ampicillin (100 mg/mL) and tetracycline (12.5 mg/mL), then plated onto NGM plates containing 100 mg/mL ampicillin and 3 mM isopropyl β-D-thiogalactoside (IPTG) (RNAi plates). RNAi-expressing bacteria were allowed to grow overnight at 37 °C. Gravid adults were transferred to RNAi-expressing bacterial lawns and allowed to lay eggs for 2 hours. The gravid adults were then removed, and the eggs were allowed to develop at 20 °C in young adults for subsequent assays. First-generation RNAi was performed from eggs to young adults for all genes, except for *elt-2* RNAi, which was conducted after animals reached the L4 stage on nematode growth medium.*unc-22* RNAi was included as a positive control to account for RNAi efficiency. All RNAi clones were from the Ahringer RNAi library. Note: Neuronal RNAi is inefficient by feeding; the results here indicate that classic immune pathways in non-neuronal tissues are not required for *E. faecium*-mediated neuroprotection.

## RNA sequencing and analysis

Approximately 100 gravid animals were placed on 10-cm NGM plates seeded with *E. coli* OP50 for 3 hours to obtain a synchronized population, which developed to the young adult stage at 20 °C. The synchronized young adult worms were washed three times in M9 buffer and transferred to *E. faecium* lawns grown on 2% agar BHI plates for colonization for 24 hours at 25 °C. The animals were then washed off the plates with M9, frozen in QIAzol using ethanol/dry ice, and stored at −80 °C prior to RNA extraction. Total RNA was extracted using the RNeasy Plus Universal Kit (Qiagen, the Netherlands), and residual genomic DNA was removed using TURBO DNase (Life Technologies, Carlsbad, CA). A total of 6 µg of RNA was reverse-transcribed with random primers using the High-Capacity cDNA Reverse Transcription Kit (Applied Biosystems, Foster City, CA).

Library construction and RNA sequencing on the BGISEQ-500 platform were performed according to established protocols, and paired-end reads of 100 bp were obtained for subsequent data analysis. The RNA sequence data were analyzed using a workflow constructed for Galaxy (https://usegalaxy.org). The RNA reads were aligned to the *C. elegans* genome (WS230) using the aligner STAR. Counts were normalized for sequencing depth and RNA composition across all samples. Differential gene expression analysis was performed using normalized samples, and genes exhibiting at least a 2-fold change were considered differentially expressed. The differentially expressed genes were subjected to SimpleMine tools from WormBase to obtain information such as WormBase IDs and gene names for further analyses. GO analysis was performed using the WormBase IDs in the DAVID Bioinformatics Database and a *C. elegans* data enrichment analysis tool.

## RNA isolation and quantitative reverse transcription PCR (qRT-PCR)

Animals were synchronized, and total RNA was extracted as described. qRT-PCR was performed using the Applied Biosystems One-Step Real-Time PCR protocol with SYBR Green fluorescence on an Applied Biosystems 7900HT machine. Reactions were analyzed per the manufacturer's instructions. Relative fold changes were calculated using the comparative CT ($2^{-\Delta\Delta CT}$) method and normalized to pan-actin (*act-1, -3, -4*). Cycle thresholds were determined using StepOnePlus Real-Time PCR System Software v2.3. All results are based on three independent experiments, each performed in triplicate, with primer sequences listed in S4 Table.

## Measurement of reactive oxygen species (ROS) in *C. elegans*

Worms were pretreated with *E. faecium* and exposed to paraquat, then subjected to ROS assays. ROS formation was measured using the fluorescent probes 2′,7′-dichlorofluorescein diacetate (H2DCFDA) (Sigma-Aldrich

D6883). Animals with 24 hours *E. faecium* pretreatment and 24 hours of paraquat treatment were collected into 1 mL PBS, washed three times, and incubated with 50 µM H2DCFDA for 60 min at room temperature. Controls were incubated with DMSO. After washing three times with PBS, animals were immobilized using 30 mM sodium azide (Sigma) in M9 buffer, mounted on a 2% agarose pad, and visualized using a Leica M165 FC fluorescence stereomicroscope. DCF fluorescence intensity was measured at excitation/emission wavelengths of 488/510–560 nm using laser scanning confocal microscopy. The assay was performed in three independent trials with at least 25 animals per assay.

## COPAS biosorter GFP analysis

DCF fluorescence intensity was analyzed using the COPAS Biosort flow cytometer (Union Biometrica). Worms pretreated with *E. faecium* and exposed to paraquat were washed three times and incubated with 50 µM H2DCF-DA before analysis. Fluorescence levels were measured using a 405-nm excitation laser and a 405-nm band-pass excitation filter with a neutral density of 0.6. The emission data were collected using a 512/25-nm band-pass emission filter (green channel). The acquired data were exported into Excel files and analyzed using Prism 9 (GraphPad).

## Intestinal rescue of NHR-86

The *vha-6* promoter and *nhr-86* coding regions were PCR-amplified separately from Bristol N2 *C. elegans* genomic DNA using the primers listed in S4 Table. The two fragments were fused by overlapping PCR to generate the *vha-6p*::*nhr-86* sequence. This sequence and the backbone plasmid pMS04 (Addgene #168172) were digested with the *Kpn* I and *Bgl* II restriction enzymes. The fused *vha-6p*::*nhr-86* fragment was then ligated into plasmid pMS04 between the *Kpn* I and *Bgl* II sites to create the expression construct pMS04_*vha-6p*::*nhr-86*::*mCherry*. The final plasmid was purified and verified by sequencing. Young adult hermaphrodite *sod-1(tm776) II; unc-119(+) III; rtSi008 [sod-1p::sod-1A4V M::sod-1 3'UTR + Cbr-unc-119(+)] IV; vsIs48 [unc-17p::GFP]; nhr-86(tm2590)* animals were transformed by microinjection of plasmids into the gonads. A mixture containing the pMS04_*vha-6p*::*nhr-86*::*mCherry* plasmids (25 ng/µL) and pRF4_ *rol-6(su1006)* (25 ng/µL) as a transformation marker was injected into the worm. Successful transformation was determined by the identification of the selection marker as a dominant roller. At least three independent lines carrying extrachromosomal arrays were obtained for each construct.

## Statistical analysis

Data were analyzed using a two-tailed Student *t* test for independent samples. For comparisons involving more than two groups, one-way ANOVA followed by post hoc analysis was employed. All experiments were conducted at least three times, with error bars representing the standard deviation unless otherwise noted. Statistical significance was defined as $P < 0.05$. The number of animals per experiment is denoted by "n". "NS" indicates non-significant results, while asterisks denote significance levels: * $P < 0.05$, ** $P < 0.01$, *** $P < 0.001$. Survival fractions were calculated using the Kaplan–Meier method, and differences between survival curves were assessed with the log-rank test (S5 Table).

## Supporting information

**S1 Fig. Motor neuron degeneration under paraquat-induced oxidative stress.** Representative images show animals lacking **(A)** zero, **(B)** one, **(C)** two, and **(D)** more than 2 motor neurons posterior to the vulva. Animals missing at least two neurons were scored as defective. Missing neurons are marked with red circles.
(PDF)

**S2 Fig. Motor neuron degeneration with or without *Enterococcus faecium* pretreatment.** Motor neuron degeneration in *sod-1* WT^M and in *sod-1* A4V^M animals was assessed under paraquat-induced oxidative stress, with or without

*E. faecium* pretreatment. Animals missing at least two neurons were scored as defective. A two-tailed Student *t* test was performed to compare data between different treatments within the same group.
(PDF)

**S3 Fig. Live *Enterococcus faecium* is required for neuroprotection. (A)** *sod-1* A4V$^M$ animals were pretreated for 24 hours with live, UV-killed, or heat-killed *E. faecium* before exposure to paraquat-induced oxidative stress. Cholinergic motor neuron integrity was assessed. Data are represented as mean±SD from three independent experiments. Statistical significance was determined using one-way ANOVA with Tukey's multiple comparison test (***$p < 0.001$). **(B)** Summary of motor neuron defects from (A) across three independent experiments.
(PDF)

**S4 Fig. Oxidative stress-induced glutamatergic neuron degeneration in *sod-1* G85R$^C$ animals. (A)** Representative images of *sod-1* G85R$^C$ animals showing intact glutamatergic sensory neurons in the tail labeled with DiD. After paraquat treatment, PHA and PHB neurons fail to take up the dye, indicating degeneration or neuronal loss. Scale bar represents 10 μm. **(B)** Percentage of defective PHA and PHB neurons in *sod-1* G85R$^C$ ALS model animals after paraquat treatment. Results from three independent trials are shown. Error bars indicate ±SD. Statistical significance was determined using one-way ANOVA with Tukey's multiple comparison test (***$p < 0.001$).
(PDF)

**S5 Fig. Enrichment analysis of *Enterococcus faecium* regulated genes.** Enrichment analysis of **(A)** *E. faecium* upregulated genes and **(B)** *E. faecium* downregulated genes using a Wormbase enrichment analysis tool (https://wormbase.org/tools/enrichment/tea/tea.cgi).
(PDF)

**S6 Fig. Motor neuron degeneration under different RNAi knockdowns.** *sod-1* A4V$^M$ animals were fed control, *pmk-1*, *elt-2*, *daf-16*, or *skn-1* RNAi. Motor neuron degeneration was assessed under paraquat-induced oxidative stress, with or without *Enterococcus faecium* pretreatment. Animals missing at least two neurons were scored as defective. A two-tailed Student *t* test was performed to compare data between different treatments within the same group.
(PDF)

**S7 Fig. mRNA levels of cytochrome P450 genes.** qRT-PCR validation of *cyp* genes that were up-regulated with *Enterococcus faecium* pretreatment ($N = 3$ biological replicates). Values are expressed as the fold difference compared with *sod-1* WT$^M$ animals fed on *Escherichia coli* ±SD by one-way ANOVA with Tukey's multiple comparison test.
(PDF)

**S8 Fig. Motor neuron degeneration under *cyp-35A12345* RNAi.** *sod-1* A4V$^M$ animals were fed control or *cyp-35A12345* RNAi. Motor neuron degeneration was assessed under paraquat-induced oxidative stress, with or without *Enterococcus faecium* pretreatment. Animals missing at least two neurons were scored as defective. One-way ANOVA was performed to compare the data. P values: Empty vector, *Ec*-paraquat versus Empty vector, *Ef*-paraquat, $P < 0.0001$; Empty vector, *Ec*-paraquat versus *cyp-35A12345* RNAi, *Ec*-paraquat, $P = 0.9997$; Empty vector, *Ef*-paraquat versus *cyp-35A12345* RNAi, *Ef*-paraquat, $P < 0.0001$; *cyp-35A12345* RNAi, Ec-paraquat versus *cyp-35A12345* RNAi, *Ef*-paraquat, $P = 0.0022$.
(PDF)

**S9 Fig. Screen of NHRs required for *cyp* activation by *Enterococcus faecium*. (A)** Screening of *nhr* genes using **(A)** *cyp-35A3* and **(B)** *cyp-35A5* as reporters. Young adult *nhr* mutant animals pretreated on *E. faecium* or *Escherichia coli* lawns were harvested for RNA extraction and qRT-PCR assay. ($N = 3$ biological replicates). Values represent fold

differences relative to animals fed on *E. coli*, analyzed by one-way ANOVA with Tukey's multiple comparisons test, * $P < 0.05$, ** $P < 0.01$, *** $P < 0.001$.
(PDF)

**S10 Fig. Motor neuron degeneration under *nhr-86* or *nhr-203* RNAi.** *sod-1* A4V<sup>M</sup> animals were fed control, *nhr-86* or *nhr-203* RNAi. Motor neuron degeneration was assessed under paraquat-induced oxidative stress, with or without *Enterococcus faecium* pretreatment. Animals missing at least two neurons were scored as defective. One-way ANOVA was performed to compare the mean of *nhr-86* RNAi and *nhr-203* RNAi with the mean of empty vector pretreated with *E. faecium*.
(PDF)

**S11 Fig. Quantification of DCF fluorescence intensity.** Intracellular ROS analysis of *sod-1* A4V<sup>M</sup> animals after *cyp-35A12345* knockdown, following *Enterococcus faecium* pretreatment and paraquat exposure. $n = 25$. The "$n$" represents the number of animals in each experiment. $N = 3$ biological replicates.
(PDF)

**S12 Fig. Intracellular ROS analysis of *sod-1* A4V<sup>M</sup> animals. (A)** Dot-plot representation of green fluorescence intensity versus TOF of empty vector and *nhr-203* RNAi animals exposed to *Enterococcus faecium* or *Escherichia coli* followed by paraquat exposure ($n = 300$). **(E)** Frequency distribution of green fluorescence in vector or *nhr-203* RNAi animals exposed to *E. faecium* or *E. coli* followed by paraquat exposure. Three independent experiments were conducted. Empty vector*, Ec*-paraquat versus Empty vector*, Ef*-paraquat, $P = $NS; *nhr-203* RNAi*, Ec*-paraquat versus *nhr-203* RNAi*, Ef*-paraquat, $P = $NS.
(PDF)

**S13 Fig. mRNA levels of *nhr-86*.** qRT-PCR test *nhr-86* expression *Enterococcus faecium* pretreatment ($N = 3$ biological replicates). Values are expressed as the fold difference compared with *sod-1* WT<sup>M</sup> animals fed on *Escherichia coli* ±SD by one-way ANOVA with Tukey's multiple.
(PDF)

**S14 Fig. mRNA levels of *cyp* genes with *nhr-86* rescue** . mRNA level of **(A)** *cyp-35A3* and **(B)** *cyp-35A5* in *sod-1* A4V<sup>M</sup>, *sod-1* A4V<sup>M</sup>;*nhr-86(tm2590)*, and *sod-1* A4V<sup>M</sup>;*nhr-86(tm2590);nhr-86p*::*nhr-86* animals. ($N = 3$ biological replicates). Values are expressed as the fold difference compared with *sod-1* A4V<sup>M</sup> animals treated by *Escherichia coli* ±SD by one-way ANOVA with Tukey's multiple comparisons test, * $P < 0.05$, ** $P < 0.01$, *** $P < 0.001$.
(PDF)

**S15 Fig. Motor neuron degeneration under different RNAi knockdowns. (A)** Oxidative stress induced cholinergic motor neuron degeneration in *sod-1* A4V<sup>M</sup> animals. *sod-1* A4V<sup>M</sup>;*sid-1(qt9)*; *rgef-1p*::GFP;*rgef-1p*::*sid-1* animals (AA258) were fed control, *nhr-86*, *dnc-1*, or *smn-1* RNAi. After growing to young adults on RNAi plates, the animals were treated with *Enterococcus faecium* or *Escherichia coli* for 24 hours before a 24-hour paraquat exposure. Cholinergic motor neuron integrity was accessed by scoring for the absence of at least two cholinergic motor neurons posterior to the vulva. * $P < 0.05$, ** $P < 0.01$, *** $P < 0.001$, two-way ANOVA. **(B)** Summary of motor neuron defects from (A) across three independent experiments.
(PDF)

**S16 Fig. mRNA levels of *cyp* genes with *nhr-86* rescue in intestine.** mRNA level of **(A)** *cyp-35A3* and **(B)** *cyp-35A5* in *sod-1* A4V<sup>M</sup>, *sod-1* A4V<sup>M</sup>;*nhr-86(tm2590)*, and *sod-1* A4V<sup>M</sup>;*nhr-86(tm2590);vha-6p*::*nhr-86* animals. ($N = 3$ biological replicates). Values are expressed as the fold difference compared with *sod-1* A4V<sup>M</sup> animals treated by *Escherichia. coli* ±SD by one-way ANOVA with Tukey's multiple comparisons test, * $P < 0.05$, ** $P < 0.01$, *** $P < 0.001$.
(PDF)

**S1 Table. Upregulated and downregulated genes upon exposure to *Enterococcus faecium.***
(XLSX)

**S2 Table. GO enrichment analysis.**
(XLSX)

**S3 Table. Strains used in the study.**
(XLSX)

**S4 Table. Primers used in the study.**
(XLSX)

**S5 Table. Mean survival under different treatment.**
(XLSX)

**S1 Data. Numerical data underlying the figures.**
(XLSX)

## Acknowledgments

Most strains used in this study were obtained from the *Caenorhabditis* Genetics Center (CGC), which is funded by the NIH Office of Research Infrastructure Programs (P40 OD010440) and the National BioResource Project (NBRP) of Japan.

## Author contributions

**Conceptualization:** Yu Sang, Alejandro Aballay.

**Data curation:** Yu Sang, Jie Ren.

**Formal analysis:** Yu Sang.

**Funding acquisition:** Alejandro Aballay.

**Methodology:** Yu Sang, Jie Ren.

**Project administration:** Alejandro Aballay.

**Software:** Yu Sang.

**Supervision:** Alejandro Aballay.

**Validation:** Jie Ren.

**Writing – original draft:** Yu Sang, Jie Ren, Alejandro Aballay.

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
