## [Editor Report · Decision Letter 0]

1 Jun 2025

Dear Yu,

Thank you for submitting your manuscript entitled "Probiotic E. faecium Protects Motor Neurons from Oxidative Stress via Gut-Activated NHR-86–Cytochrome P450 Pathway" for consideration as a Research Article by PLOS Biology.

Your manuscript has now been evaluated by the PLOS Biology editorial staff, as well as by an academic editor with relevant expertise, and I am writing to let you know that we would like to send your submission out for external peer review.

Once your full submission is complete, your paper will undergo a series of checks in preparation for peer review. After your manuscript has passed the checks it will be sent out for review. To provide the metadata for your submission, please Login to Editorial Manager (https://www.editorialmanager.com/pbiology) within two working days, i.e. by Jun 03 2025 11:59PM.

Kind regards,

Melissa

Melissa Vazquez Hernandez, Ph.D.

Associate Editor

PLOS Biology

---

## [Decision Letter · Decision Letter 1]

15 Jul 2025

Dear Dr Sang,

Thank you for your patience while your manuscript "Probiotic E. faecium Protects Motor Neurons from Oxidative Stress via Gut-Activated NHR-86–Cytochrome P450 Pathway" was peer-reviewed at PLOS Biology. Your manuscript has been evaluated by the PLOS Biology editors, an Academic Editor with relevant expertise, and by two independent reviewers.

As you will see in the reviewer reports, although the reviewers acknowledge the potential interest in your findings, they have also raised a substantial number of crucial concerns. Based on their specific comments and following discussion with the Academic Editor, it is clear that a substantial amount of work would be required to meet the criteria for publication in PLOS Biology. Given our and the reviewer interest in your study, we would be open to inviting a comprehensive revision of the work; however, this would need to thoroughly address all the reviewers' comments, specially regarding deeper mechanistic insights. Reviewer 1 thinks that identifying NHR-86 as a key factor is interesting but the study lacks enough mechanistic insights. The reviewer gives multiple experimental suggestions like measuring if nhr-86 expression is regulated by E. faecium, how intestinal activation of NHR-86 confer protection to neurons, to evaluate if overexpression of nhr-86 itself confers protection, transcriptomics on nhr-86 mutants, to show if E. faecium needs to be alive to confer protection, among others. Reviewer 2 thinks the study would benefit on addressing the generality of E. faecium suppression in other ALS models and to determine if catecholaminergic signaling is implicated in the communication. This reviewer also mentions that how oxidative response in the intestine is connected to neurodegeneration of motor neurons is not experimentally addressed.

Given the extent of revision that would be needed, we cannot make a decision about publication until we have seen the revised manuscript and your response to the reviewers' comments. Your revised manuscript would need to be seen by the reviewers again, but please note that we would not engage them unless their main concerns have been addressed.

We appreciate that these requests represent a great deal of extra work, and we are willing to relax our standard revision time to allow you 6 months to revise your study. Please email us (plosbiology@plos.org) if you have any questions or concerns, or envision needing a (short) extension.

**IMPORTANT - SUBMITTING YOUR REVISION**

*Resubmission Checklist*

*Published Peer Review*

*PLOS Data Policy*

*Blot and Gel Data Policy*

Sincerely,

Melissa

Melissa Vazquez Hernandez, Ph.D.

Associate Editor

PLOS Biology

REVIEWERS' COMMENTS:

Reviewer #1:

In this study, Sang et al., found that the probiotic E. faecium appears to be protect motor neuron degeneration through NHR-86, which activates the expression of cytochrome P450, which reduces intracellular levels of ROS to reduce oxidative stress and confer neuroprotection. Although the study is interesting in identifying NHR-86 as a key factor, it falls short in elucidating the mechanisms underlying the activation of NHR-86. Below are some comments that can hopefully help improve the manuscript.

Major

1. Is the expression of nhr-86 regulated by E. faecium? This can be easily tested by treating the nhr-86::GFP strain with E. faecium.

2. In what tissue does nhr-86 function to confer neuroprotection? The authors mentioned that nhr-86 is expressed in hypodermis, intestine, and head neurons. So, the authors can use tissue-specific promoter to drive nhr-86 expression and identify the site of action for nhr-86. One likely place is the intestine, since it appears to be where nhr-86 is expressed the strongest. If that is the case, the author may want to explain or hypothesize how intestinal activation of NHR-86 and CYP genes confer protection on motor neurons through cell-non-autonomous mechanisms.

3. One the other hand, does overexpression of nhr-86 itself confers protection against motor neuron degeneration under E. coli diet? This is a crucial experiment, because it can test whether the expression level of nhr-86 itself is sufficient for neuroprotection or it must be activated by some ligands present in the probiotic E. faecium.

4. Is there any transcriptomic data on nhr-86 mutants? If available, the authors can compare the NHR-86-regulated genes with E. faecium-regulated genes to see if NHR-86 is indeed the major effector downstream of E. faecium. A simple search of the literature led to this paper (PMID: 30668573). It seems Peterson et al., has even done ChIP-Seq on NHR-86. These data gave the possibility for the above analysis. The authors can also look at the CYP genes in the ChIP-seq data to see if they are bound by NHR-86.

5. Can E. faecium colonize the gut of C. elegans? This can be tested through smFISH staining like previously performed (PMID: 32669368). Can UV-killed E. faecium still confer protection. It will be important to know whether E. faecium needs to be alive and colonize the gut of C. elegans to confer protection or it can simply be served as a diet to confer protection.

6. In terms of the disease modeling, does SOD-1 A4V mutants show motor neuron loss or defects without paraquat treatment?

7. From Figure 1C, it seems that if SOD-1 is WT, E. faecium does not confer protection. So, the SOD1 A4V mutants is required to see the effects of E. faecium. Since the authors are trying to model ALS using the mutant, they must at least discuss how the SOD mutation contributes to the hypersensitivity to oxidative stress and how it interacts with E. faecium. For example, is the E. faecium-induced CYP upregulation not sufficient to counter the oxidative stress in the SOD WT condition? Or is it because the phenotype is not severe enough to see any difference?

8. "Since the innate immune response cluster is among the most down-regulated, we speculated that knocking down immune pathways would decrease E. faecium-mediated protection against motor neuron degeneration." The logic of this sentence is flawed. If innate immune genes are downregulated by E. faecium, that most likely means the innate immune genes are neurotoxic, so downregulating them can facilitate neuroprotection. So, the prediction is that overexpression of these innate immune gene would block E. faecium-mediated neuroprotection. So, the experiments in Figure 3A does not mean much. I will suggest overexpressing these genes to test their effects.

Minor

1. What promoter was used to drive the expression of sod-1 WT and A4V? This is not very clear to me. Is it expressed in all motor neurons?

2. Figure 1A, I suggest adding some labels on the images to show which one is Ec-treated and which ones are Ef-treated.

3. When examining neuronal loss, did the author notice whether the number of neurons that are defective are different between Ec and Ef-treated animals (when they are classified as animals with neurons defective)?

4. For Figure 5D and E, there need to be some statistical analysis on the results.

5. "We performed an RNAi screen targeting 13 nhr genes responsible for regulating cyp genes" Judging from Figure 5A, this is not a RNAi screen but rather a candidate mutant screen.

Reviewer #2:

Summary

This manuscript reports that exposure to E. faecium suppresses oxidative damage in a C. elegans SOD1 ALS model with results from RNA sequencing and other studies that implicate nuclear hormone receptor NHR-86 regulation of cyp gene expression in the intestine as a key player in this process. The studies are undertaken with appropriate controls, the conclusions are reasonable, and the manuscript is of high-quality with admirably clear descriptions. There are a few points that are missing in the text (noted below).

The manuscript would have much more impact in the ALS or gut/brain axis field if the authors included studies that 1) address the generality of E. faecium suppression in other ALS or SOD1 ALS models and/or 2) determine if catecholaminergic signaling is implicated in communication between the intestine and the gut. Without these, the impact of the paper is somewhat diminished, but still of interest to researchers interested in oxidative stress response.

Major comments

Evidence that exposure to E. faecium suppresses sod-1A4V motor neuron defects is convincing. But in ALS patients, including SOD1A4V patients, glutamatergic neurons are also lost. Does the sod-1A4V model also show glutamatergic neuron loss and is this suppressed by E. faecium? Is there another model that allows assessment of glutamatergic neurons? Probing this would considerably increase the impact of the manuscript.

C. elegans SOD1 overexpression models lead to degeneration of neurons, albeit not cholinergic motor neurons. Because degeneration in over-expression models does not require pre-treatment with oxidative stressors, these should be examined to determine the general applicability of the protection provided by E. faecium pre-treatment.

The studies shown in Figures 3 and 5 used RNAi by feeding to determine if knockdown of transcription factors decreased E. faecium neuroprotection. Most readers will not know that C. elegans neurons are insensitive to RNAi by feeding and that conclusions drawn about these genes from these studies only applies to a possible role for these genes in the intestine and other non-neuronal tissues. This caveat should be explicitly mentioned in both the main text and in the figure legend.

How oxidative response in the intestine is functionally connected to neurodegeneration of motor neurons is not experimentally addressed and probing this would considerably increase the impact of the manuscript. The manuscript discussion includes a discussion paragraph suggesting two possible mechanisms, citing general reviews in the field of gut/brain axis: (1) microbial-derived short-chain fatty acids and (2) catecholamine neurotransmission changes. The latter should be straightforward to test experimentally using C. elegans mutant strains or RNAi, especially as the signal may be arising from the intestine.

Minor Comments

Does exposure to E. faecium protect in nonSOD1 ALS or FTD models? It's not critical to include this, but readers will ask this obvious question.

The Introduction states that "Over forty gene mutations have been linked to ALS, with the most prevalent occurring in the sod-1 gene…", but the fraction of ALS cases caused SOD1 mutations is very low compared to other genetic causes of ALS. Correction and citation adding another citation would be helpful.

If C. elegans genes/pathways have been previously implicated in intestinal protection of C. elegans neurons, then those should also be cited/discussed in the discussion section.

The Methods section states that "All samples were run in triplicate". Were three independent biological replicates used? Please clarify for the reader.

Were researchers blinded as to genotype/treatment when capturing/selecting images for analysis or for any scoring that was undertaken manually? An explicit statement in the Methods section is needed.

---

## [Decision Letter · Decision Letter 2]

9 Jan 2026

Dear Yu,

Thank you for your patience while we considered your revised manuscript "Probiotic E. faecium Protects Motor Neurons from Oxidative Stress via Gut-Activated NHR-86–Cytochrome P450 Pathway" for publication as a Research Article at PLOS Biology. This revised version of your manuscript has been evaluated by the PLOS Biology editors, the Academic Editor and the original reviewers.

Based on the reviews, provided you satisfactorily address the remaining editorial points raised by the reviewers. Please also make sure to address the following data and other policy-related requests.

1) We routinely suggest changes to titles to ensure maximum accessibility for a broad, non-specialist readership, and to ensure they reflect the contents of the paper. In this case, we would suggest a minor edit to the title, as follows. Please ensure you change both the manuscript file and the online submission system, as they need to match for final acceptance:

"A gut-activated NHR-86–CYP pathway mediates the neuroprotective effects of Enterococcus faecium probiotics in a nematode model of amyotrophic lateral sclerosis"

Please supply the numerical values either in the a supplementary file or as a permanent DOI’d deposition for the following figures:

Figure 1CDE, 2ABC, 3B, 4ABCE, 5ABDEFG, 6B-G, 7B-E, S2, S3AB, S4B, S5AB, S6, S7, S8, S9AB, S10, S11, S12AB, S13, S14AB, S15AB, S16AB

3) Please cite the location of the data clearly in all relevant main and supplementary Figure legends, e.g. “The data underlying this Figure can be found in S1 Data” or “The data underlying this Figure can be found in https://doi.org/10.5281/zenodo.XXXXX”

4) Please ensure that you are using best practice for statistical reporting and data presentation. These are our guidelines https://journals.plos.org/plosbiology/s/best-practices-in-research-reporting#loc-statistical-reporting and a useful resource on data presentation https://journals.plos.org/plosbiology/article?id=10.1371/journal.pbio.1002128

-- If you are reporting experiments where n ≤ 5, please plot each individual data point.

5) Supplementary files (e.g., excel). Please ensure that all data files are uploaded as 'Supporting Information' and are invariably referred to (in the manuscript, figure legends, and the Description field when uploading your files) using the following format verbatim: S1 Data, S2 Data, etc. Multiple panels of a single or even several figures can be included as multiple sheets in one excel file that is saved using exactly the following convention: S1_Data.xlsx (using an underscore).

6) Please ensure that your Data Statement in the submission system accurately describes where your data can be found and is in final format, as it will be published as written there

7) Per journal policy, if you have generated any custom code during the course of this investigation, please make it available without restrictions. Please ensure that the code is sufficiently well documented and reusable, and that your Data Statement in the Editorial Manager submission system accurately describes where your code can be found. More information on our Code Policy, what and how to share can be found here: https://journals.plos.org/plosbiology/s/code-availability

We expect to receive your revised manuscript within two weeks.

*Published Peer Review History*

*Press*

Sincerely,

Melissa

Melissa Vazquez Hernandez, Ph.D.

Associate Editor

PLOS Biology

REVIEWERS' COMMENTS

REVIEWERS' COMMENTS

Reviewer #1: The authors have done a good job addressing my concerns. I support the publication of the paper.

Reviewer #2: The authors have nicely addressed all of my comments

---

## [Editor Report · Decision Letter 3]

16 Jan 2026

Dear Yu,

Thank you for the submission of your revised Research Article "A gut-activated NHR-86–CYP pathway mediates the neuroprotective effects of Enterococcus faecium probiotics in a nematode model of amyotrophic lateral sclerosis" for publication in PLOS Biology. On behalf of my colleagues and the Academic Editor, Ursula Jakob, I am pleased to say that we can in principle accept your manuscript for publication, provided you address any remaining formatting and reporting issues. These will be detailed in an email you should receive within 2-3 business days from our colleagues in the journal operations team; no action is required from you until then. Please note that we will not be able to formally accept your manuscript and schedule it for publication until you have completed any requested changes.

PRESS

Sincerely,

Melissa

Melissa Vazquez Hernandez, Ph.D., Ph.D.

Associate Editor

PLOS Biology
